# Phenotypes Associated with Pathogenicity: Their Expression in Arctic Fungal Isolates

**DOI:** 10.3390/microorganisms7120600

**Published:** 2019-11-22

**Authors:** Laura Perini, Diana C. Mogrovejo, Rok Tomazin, Cene Gostinčar, Florian H. H. Brill, Nina Gunde-Cimerman

**Affiliations:** 1Department of Biology, Biotechnical Faculty, University of Ljubljana, Jamnikarjeva 101, SI-1000 Ljubljana, Slovenia; cene.gostincar@bf.uni-lj.si (C.G.); Nina.Gunde-Cimerman@bf.uni-lj.si (N.G.-C.); 2MicroArctic Research Group, Dr. Brill + Partner GmbH Institut für Hygiene und Mikrobiologie, Stiegstück 34, 22339 Hamburg, Germany; carito191286@hotmail.com (D.C.M.); florian.b@brillhygiene.com (F.H.H.B.); 3Institute of Microbiology and Immunology, Faculty of Medicine, University of Ljubljana, Zaloška 4, SI-1000 Ljubljana, Slovenia; Rok.Tomazin@mf.uni-lj.si; 4Lars Bolund Institute of Regenerative Medicine, BGI-Qingdao, Qingdao 266555, China

**Keywords:** Arctic, fungi, emerging pathogens, thermotolerance, hemolysis, antifungal resistance

## Abstract

Around 85% of the environments on Earth are permanently or seasonally colder than 5 °C. Among those, the poles constitute unique biomes, which harbor a broad variety of microbial life, including an abundance of fungi. Many fungi have an outstanding ability to withstand extreme conditions and play vital ecosystem roles of decomposers as well as obligate or facultative symbionts of many other organisms. Due to their dispersal capabilities, microorganisms from cryosphere samples can be distributed around the world. Such dispersal involves both species with undefined pathogenicity and potentially pathogenic strains. Here we describe the isolation of fungal species from pristine Arctic locations in Greenland and Svalbard and the testing of the expression of characteristics usually associated with pathogenic species, such as growth at 37 °C, hemolytic ability, and susceptibility to antifungal agents. A total of 320 fungal isolates were obtained, and 24 of the most abundant and representative species were further analyzed. Species known as emerging pathogens, like *Aureobasidium melanogenum*, *Naganishia albida*, and *Rhodotorula mucilaginosa*, were able to grow at 37 °C, showed beta-hemolytic activity, and were intrinsically resistant to commonly used antifungals such as azoles and echinocandins. Antifungal resistance screening revealed a low susceptibility to voriconazole in *N. albida* and *Penicillium* spp. and to fluconazole in *Glaciozyma watsonii* and *Glaciozyma*-related taxon.

## 1. Introduction

Despite appearing devoid of life, the poles are now recognized as biomes [1], and it has been repeatedly demonstrated that these unique environments are colonized by a wide variety of microbial life [2]. In addition to bacteria, a great diversity of psychrophilic, as well as psychrotolerant fungi inhabit the cryosphere, where they play important roles in the ecosystem [3]. For example, a strong association between the abundance and diversity of fungi and the blooming of algae on the dark surface ice of the Greenland Ice Sheet has been described [4]. Particularly common are species of the genera *Penicillium*, *Thelebolus*, *Alternaria*, *Rhodotorula*, *Vishniacozyma*, *Aspergillus*, and *Cladosporium* [3,4,5,6].

Due to their small size, microorganisms generally disperse over large distances, decreasing the impact of geographical barriers [7,8]. At least for some species, a constant exchange of populations between fairly remote habitats is possible [9]. For fungi, and in particular those inhabiting the poles, the main dispersal vector is the air [10]. Melting glaciers and, subsequently, ocean currents also contribute to the dispersal [11]. Biological particles and bioaerosols (bacterial and fungal) account for more than 20% of the total atmospheric particle content [12,13]. Moreover, microorganisms characteristic of temperate and warm climates, including some plant and animal pathogens, are abundantly present in the cryosphere, from where they can be transported and established ubiquitously around the world [12].

Fungi can efficiently adapt to new conditions and environments. High adaptability and stress tolerance often correlate with increased potential for causing opportunistic infections [14]. However, the pathogenicity potential of polar fungi is very poorly understood, although they might include potentially pathogenic strains [15]. Nutrient limitation, UV radiation, pH, osmotic and oxidative stresses are only some of the conditions that fungi must overcome to colonize and thrive in extreme Arctic environments. Such adaptations might increase their capacity to survive unexpected habitat shifts, including transitions into habitats associated with humans [16,17]. In fact, some polyextremotolerant fungi (as long as they can grow at elevated temperatures) have the potential to colonize complex organisms (plants or invertebrates) and even warm-blooded animals [14,18].

While fungal infections are on the rise, the treatment options remain limited and are being further decreased by the rise of antimicrobial resistance. Due to the limited number of physiological differences between animal and fungal cells (both being eukaryotes), few antifungal agents with low animal toxicity are currently available. Azoles act as inhibitors of ergosterol production and as such have very broad-spectrum activity and relatively low mammalian cell toxicity. They are the most common clinically used antifungal agents and can be classified, in terms of structure, into two categories: imidazoles and triazoles [19]. Among the imidazoles, only ketoconazole, one of the first developed azoles, has systemic activity. Because of its severe side effects and high gastric and hepatic toxicity, ketoconazole has been largely replaced by the first generation of triazoles for the treatment of systemic infections [19]. A representative of the triazole class is fluconazole, which is active against many species, except for opportunistic filamentous fungi, including *Aspergillus*, *Fusarium*, and *Mucorales* [20]. Because of its low toxicity, excellent pharmacological characteristics, and antiyeast activity, it has a very important role in the treatment of candidiasis and cryptococcosis [21,22]. Both imidazoles and the first generation of triazoles have a fungistatic nature. In order to overcome and widen the therapeutic spectrum with the antimold activity, a second generation of triazoles was developed. Second-generation triazoles, such as voriconazole, are considered fungicidal [23]. Echinocandins are the newest antifungal agents and the only ones that affect the fungal cell wall, acting as inhibitors of the β-(1,3)-d-glucan synthesis. All three echinocandins, caspofungin, anidulafungin, and micafungin, are lipopeptides derived from fungal natural products. Depending on the target microorganism and their concentration, echinocandins have both fungistatic (*Aspergillus* spp.—low concentrations) and fungicidal (*Candida* spp.—high concentrations) activity [20].

Antifungal resistance is the result of the natural selection of microorganisms with the ability to survive and thrive in the presence of drugs [20]. In contrast to the rapid emergence and spread of acquired resistance to antimicrobials that is characteristic of bacteria, fungi develop resistance slowly, gradually alternating physiological functions. Thus, the problem is not so much acquired resistance (as is the case with bacteria), but the emergence of new, opportunistic pathogens with intrinsic resistance to established antifungals [19,24]. Frequently encountered fungal pathogens such as *Candida* spp. and *Aspergillus* spp. have precisely defined clinical breakpoints (CBP) or epidemiological cutoff values (ECOFF) to help guide optimal antifungal therapy (European Committee on Antimicrobial Susceptibility Testing EUCAST, 2018). This is not the case for environmental strains and emerging pathogens, which hinders their investigation and complicates the interpretation of in vitro antifungal susceptibility testing (AFST).

While there has been considerable research about the fungal diversity of the Arctic environments [3,6,25], the potential pathogenicity of members of these fungal communities is still poorly understood [16,26]. As climate change is set to increase the incidence of diseases [27,28], Arctic fungal strains displaying phenotypes associated with pathogenicity should be investigated. Accordingly, in the present study we investigated fungal isolates from the Arctic region for three phenotypes commonly associated with pathogenicity in humans, i.e., growth at 37 °C, expression of hemolytic activity on bovine blood agar plates, and resistance to common antifungal therapeutics.

## 2. Materials and Methods

### 2.1. Site and Sample Description

Fieldwork campaigns were conducted in the summer seasons of the years 2016–2018, on the SW coast of Greenland (Greenland Ice Sheet - GrIS, and Russell Glacier) and Svalbard (Ny-Ålesund, Svalbard, Norway). Samples of snow, supraglacial ice and water, subglacial ice, glacial meltwater, and air were obtained from the locations described in Table 1. Supraglacial ice collected on the GrIS was divided based on the visible abundance of living dark glacier algae on its surface into dark ice samples (containing high biomass of glacier algae) and clear ice samples (without visible biomass of glacier algae). Subglacial ice samples were collected from three polythermal glaciers in Svalbard: Midtre Lovénbreen, Vestre Brøggerbreen, and Pedersenbreen. Samples were collected using sterilized tools and sterile nitrile gloves to avoid contamination, and organized into sterile Whirl-Pak^®^ plastic bags or sterile plastic bottles. All the samples were processed within a few hours of their collection at the primary ice camp on the Greenland Ice Sheet and the NERC Arctic Research Station (Ny-Ålesund).

### 2.2. Fungal Cultivation and Isolation

The outer surface of subglacial and supraglacial ice was discarded after melting at room temperature. Snow and ice samples were then melted at 4 °C prior to analysis. All the samples were filtered through Milli-pore membrane filters (0.45 μm pore size) in triplicate, and filters were placed onto different media. Minimal medium (MM) and synthetic nutrient-poor agar (SNA) [29,30] were used to isolate oligotrophic fungi; dichloran rose bengal chloramphenicol agar (DRBC) [31] (Biolife, Milan, Italy) was used as a general-purpose medium; Reasoner’s 2A agar (R2A) [32] (Biolife) was used to isolate bacteria; dichloran 18% glycerol (DG18) medium [33] (Biolife, Milan, Italy) was used to obtain moderate xerophilic and xerotolerant species. To prevent bacterial growth, the media contained chloramphenicol (50 mg/L), except for R2A. Plates were incubated at 15 °C for up to two months in sterile plastic bags. All strains obtained were deposited in the Ex Culture Collection of the Infrastructural Centre Mycosmo (MRIC UL) at the Department of Biology, Biotechnical Faculty, University of Ljubljana, Slovenia.

### 2.3. Fungal Identification

Fungi were transferred onto fresh MEA (malt extract agar) or PDA (potato dextrose agar, Biolife) media for a week to obtain pure cultures. Filamentous fungi DNA was extracted following the protocol described in [4], while yeast DNA was extracted using the PrepMan Ultra reagent (Applied Biosystems, Foster City, CA, USA) according to the manufacturer instructions.

Amplification of the ITS region using the primers ITS5 and ITS4 [34] was performed for filamentous fungi, while amplification of the D1 and D2 domains of the LSU gene using NL1 and NL4 primers [35] was performed for yeasts as described in [6]. For isolates belonging to *Penicillium* and *Cladosporium* spp., additional marker genes were amplified, such as β-tubulin gene (benA) with Ben2f and Bt2b primers [36], and partial sequence of actin gene (act) with ACT-512F and ACT-783R primers [37], respectively. Subsequently, amplicons were Sanger-sequenced (Microsynth AG, Balgach, Switzerland) and the resulting sequences were aligned using MUSCLE software [38], implemented in the MEGA7 package [39] and compared against the GenBank database using the nBLAST platform. Sequences are available in the NCBI GenBank nucleotide database (Table 2).

### 2.4. Thermotolerance Assay

Fungal isolates were cultivated onto MEA and SDA (Sabouraud dextrose agar, Biolife, Milan, Italy) media and incubated at 37 °C for up to seven days to test their ability to grow at high temperatures.

### 2.5. Hemolytic Assay on Blood Agar

Fungal isolates were cultured on blood agar base (Honeywell Fluka, Charlotte, NC, USA) containing 5% of sterile bovine blood and incubated at both 15 °C and 37 °C for a week. *Candida albicans* (EXF-14635; ATTC90028) was used as a positive beta-hemolytic control [40,41]. An isolate was considered alpha-hemolytic when a green zone of discoloration was observed in the media, beta-hemolytic when a clear zone was observed denoting complete lysis of the erythrocytes of the blood agar plate, and gamma-hemolytic (not hemolytic) when there was no change noted in the media [42].

### 2.6. Antifungal Susceptibility Test (AFST)

Fungi were cultured at 15 °C for up to four days in SDA prior to the test. AFST was done using MIC-gradient strip method according to the manufacturer’s instructions (bioMérieux, Marcy-l’Étoile, France). Inoculum for the AFST was prepared by homogenizing well-isolated colonies/spores from the SDA plate in sterile 0.85% NaCl solution (with the addition of 0.05 g/L of Tween 20 for filamentous fungi) to obtain a turbidity equivalent to 0.5 McFarland standard, approximately corresponding to 1–5 × 10^8^ CFU/mL. Suspensions were spread evenly over the entire surface of RPMI-1640 agar plates supplemented with 2.0% glucose (bioMérieux, Marcy-l’Étoile, France) using a sterile cotton swab. Etest gradient strips were then applied to the agar surface. Inverted plates sealed with parafilm were incubated in sterile plastic bags at 15 °C to allow the optimal growth of the Arctic strains. Fungal growth was checked after two and seven days. The antifungals tested belonged to two different antifungal classes: azoles and echinocandins. The azoles used were voriconazole (VO—range: 32–0.002 μg/mL), ketoconazole (KE—range: 32–0.002 μg/mL), and fluconazole (FL—range: 256–0.016 μg/mL). The echinocandins used were caspofungin (CS—range: 32–0.002 μg/mL), micafungin (MYC—range: 32–0.002 μg/mL), and anidulafungin (AND—range: 32–0.002 μg/mL). Given their intrinsic resistance to echinocandins [43], basidiomycetes were not tested for that antifungal class. The reference strain, *Candida albicans* ATCC^®^ 90028™, was used as a control. MIC (minimum inhibitory concentration) values were determined and are reported in Table 3.

## 3. Results

### 3.1. Fungal Isolation and Identification

A total of 320 fungal isolates were obtained from the previously described samples after incubation [6]: 45.9% belonging to Basidiomycota (147/320), 53.8% to Ascomycota (172/320), and only 0.3% belonging to Mucoromycotina (1/320). In Greenland, dark ice was the environment with the most diverse mycobiome, hosting 23 different fungal genera and 28 species, followed by snow with 18 genera and 22 species. Unlike the other sampled environments, dark ice supported the presence of several genera that contain some common plant pathogen and endophyte species, such as *Acrodontium*, *Epicoccum*, *Bjerkandera*, and *Microdochium nivale*. In Svalbard, samples of subglacial ice collected from three different glaciers hosted, in total, 27 genera and more than 35 different species. The most commonly found species in different samples and environments were *Cladosporium* sp., *Penicillium* sp., *Venturia* sp., *Glaciozyma*-related taxon, *Mrakia* sp., *Phenoliferia* sp., and *Rhodotorula svalbardensis* pro. tem. The most commonly found and most abundant species in the different environments were selected for further tests (Table 4).

### 3.2. Thermotolerance

Isolates with detectable growth at 37 °C on MEA and SDA were *Penicillium chrysogenum* (EXF-12443), *Naganishia albida* (EXF-12581), *Rhodotorula mucilaginosa* (EXF-13607), and *Aureobasidium melanogenum* (EXF-13647). All the other strains tested were not able to grow at such a temperature.

### 3.3. Hemolytic Assay on Blood Agar

Results of the hemolytic assay on bovine blood agar are listed in Table 3. The tested fungal species showed both alpha- and beta-hemolytic phenotypes. More than half of the species (13/25) were not hemolytic at 15 °C. *Aureobasidium pullulans* and *A. melanogenum* were the only isolates that presented a clear and strong beta-hemolytic reaction at 15 °C (Figure 1), whereas the remaining strains (eight out of 25), such as *Preussia* sp., *Ph. psychrophenolica*, *Neocucurbitaria* sp., and *Oleoguttula mirabilis* displayed alpha-hemolysis. *Penicillium bialowiezense*-like had zones of both alpha- and beta-hemolysis (Figure 1), while *P. chrysogenum* showed only a (weak) alpha-hemolytic reaction. *Pseudogymnoascus* sp. presented a brownish discoloration of the agar not attributable to any of the known hemolytic reactions. *Rhodotorula mucilaginosa* was the only isolate with a beta-hemolytic reaction at 37 °C. No strains were alpha-hemolytic at this temperature.

### 3.4. Antifungal Susceptibility Test (AFST)

Antifungal susceptibility was evaluated in all the selected species and results of the test are reported in Table 3.

#### 3.4.1. Azoles

Most of the tested strains expressed relatively low susceptibility to fluconazole, with MIC_50_ = 64 μg/mL (concentration inhibiting the growth of half of the tested strains). Fluconazole proved to be most active against *Phenoliferia* spp., with MIC ≤ 0.047 μg/mL. Filamentous fungi, namely *Articulospora* sp., *Cladosporium* sp., *Comoclathris* sp., *Neocucurbitaria* sp., *Penicillium bialowiezense*-like, *P. chrysogenum*, *P. crustosum*, *Preussia* sp., *Venturia* sp., and *Pseudogymnoascus* sp., and yeasts, namely *Glaciozyma* spp., showed high MICs for fluconazole ranging from 0.25 to 265 μg/mL, appearing resistant in vitro. The filamentous *Pseudogymnoascus* sp. had the lowest MIC value (8 μg/mL) compared to the other filamentous fungi. Voriconazole, on the other hand, showed better activity against the tested isolates with MIC_50_ = 0.032 μg/mL. It was most active against *Articulospora* sp., *Dothiora* sp., *Neocucurbitaria* sp., *Glaciozyma* spp., and *Phenoliferia* sp. with MICs ≤ 0.004 μg/mL. The lowest activity was observed in the case of *Naganishia albida* with MIC = 2 μg/mL. In general, ketoconazole showed similar antifungal activity to voriconazole, with MIC_50_ = 0.032 μg/mL. The lowest ketoconazole activity was observed among *Penicillium* spp., with MICs ranging from 0.38 μg/mL to 12 μg/mL.

#### 3.4.2. Echinocandins

In general, micafungin and anidulafungin showed comparable antifungal activity, with MIC_50_ = 0.008 μg/mL for both substances. Micafungin and anidulafungin both proved most active against *Cladosporium* sp., *Dothiora* sp., *Neocucurbitaria* sp., and *Penicillium* spp., with MICs ≤ 0.008 μg/mL. In general, caspofungin had a lower antifungal activity compared to micafungin and anidulafungin, with MIC_50_ = 0.125 μg/mL.

## 4. Discussion

Arctic fungi are extremophiles able to withstand environmental conditions that are too harsh for most other organisms. Their great adaptability and stress tolerance, together with the ability of yeasts and fungal spores to spread in the atmosphere and water, and the selective pressure induced by global environmental changes, may be increasing the risks they present as emerging fungal pathogens. Exoenzyme production, resistance to antifungals, and the ability to adapt to host body temperature are some of the virulence-associated characteristics shared by pathogenic fungi [19,44,45,46]. At least one of the abovementioned virulence-associated phenotypes was expressed by several of our Arctic isolates.

Thermotolerance is a crucial virulence factor for establishing the invasive type of infection in humans and is also the trait lacking in most fungi. Typically, psychrotolerant strains able to survive at higher temperature might bear a significant pathogenic potential [15]. All the selected fungal species were isolated at an incubation temperature of 15 °C, showing a psychrotolerant or psychrophilic nature. Growth tests at higher temperature (37 °C) revealed the abilities of certain strains to adapt and grow in broad temperature ranges. *Naganishia albida*, *Rhodotorula mucilaginosa*, and *Aureobasidium melanogenum* are known polyextremotolerant and emerging pathogenic species with clinical relevance, and together with *Penicillium chrysogenum* showed their ability to grow at 37 °C despite their isolation from a very cold habitat.

The first screening of hemolytic phenotypes on blood agar in environmental Arctic fungi reported here revealed that *Aureobasidium pullulans*, *A. melanogenum* (at 15 °C) and *Rhodotorula mucilaginosa* (at 37 °C) had the strongest beta-hemolytic reaction, an ability already reported in strains of these fungi isolated from hypersaline water and subglacial ice [47]. Also, penicillia (*Penicillium bialowiezense*-like and *P. chrysogenum*) exhibited some degree of hemolysis, in agreement with other studies on airborne fungi [48,49]. *P. chrysogenum* strains from indoor air were found to produce chrysolysin, a proteinaceous hemolysin, only when incubated at 37 °C [50]. The same hemolysin was not always expressed in outdoor air strains of the same species or at lower temperatures [51]. Based on our results, no reaction was observed in *P. crustosum*, suggesting that hemolytic abilities are species- or even strain-specific. The latter statement is supported by the authors of [49], who studied airbone fungi in a Russian Arctic settlement and found that *Penicillium aurantiogriseum*, but not *P. canescens* and *P. simplicissimum*, showed strong hemolytic activity. *Rhodotorula mucilaginosa* was the only isolate with a strong beta-hemolytic activity at 37 °C, and a weaker activity at 15 °C, again highlighting the importance of temperature as a triggering factor of virulence-associated traits [50]. *Pseudogymnoascus* spp. are known to have many extracellular enzymes that can cause blood cells lysis [51]. *Neocucurbitaria* sp., *Preussia* sp., *Phenoliferia psychrophenolica*, and *Oleoguttula* sp. showed some degree of hemolysis. The knowledge of hemolysis in fungal species is extremely limited compared to bacteria. Fungal hemolysins are pore-forming exotoxins capable of lysis of red blood cells and nucleated cells [52]. Little is known about their role and mechanisms of action, but research highlighted that many well-known pathogenic fungi, such as *Aspergillus flavus* and *A. fumigatus*, as well as *Candida albicans* and *Cryptococcus neoformans*, produce several cytolysins [52]. The expression of fungal hemolysins has been proposed as a virulence factor and as a survival strategy for opportunistic fungi [53,54,55]. The iron released by the lysis of red blood cells is an essential factor in fungal growth [56]; it can facilitate hyphal infection [53] and favors the transition from fungi with a commensal lifestyle to invasive pathogens [57]. However, while fungal hemolysin expression can provide an opportunity for colonization and infection, research suggests that the compounds might have functional roles unrelated to pathogenesis, such as hyphal aggregation for fruiting body formation [58,59] and regulation of fungal growth [52]. Moreover, their functional role as a competitor in specific ecological niches [60] and plant pathogenesis has also been proposed [61]. Due to the limited nutrient sources in the Arctic, hemolysins could also provide an ecological advantage to fungi, which are well-known decomposers, and help with obtaining limiting nutrients by disrupting the membranes of other microorganisms and thus accessing their intracellular content.

*Naganishia albida,* formerly *Cryptococcus albidus*, is, together with *Papiliotrema laurentii*, formerly *Cryptococcus laurentii*, responsible for about 80% of non-neoformans cryptococcal infections [62,63]. Despite the lack of validated standard antifungal treatment protocols for non-neoformans cryptococcosis, the literature supports the use of fluconazole for consolidation and maintenance therapy [22]. Our results showed that fluconazole and voriconazole were only weakly active against *N. albida* EXF-12581 in vitro (MIC = 64 μg/mL and MIC = 2 μg/mL, respectively). Ketoconazole was the only azole that showed higher in vitro susceptibility (MIC = 0.047 μg/mL). This yeast is, like all *Cryptococcus*-like fungi, intrinsically resistant to the echinocandin class.

*Filobasidium magnum*, formerly *Cryptococcus magnus*, has rarely been isolated from mammals, with only four case reports [64,65,66]. Compared to other AFST data, it is in vitro susceptible to voriconazole with an at least two 2-fold dilution steps lower MIC [46]. According to other studies on environmental isolates [67,68], *Naganishia albida* and *Filobasidium magnum* are known to be fluconazole-resistant.

Among the isolates that were able to grow at 37 °C was *Rhodotorula mucilaginosa*, a basidiomycetous yeast that emerged as an opportunistic pathogen infecting different body sites of patients with immune system deficiencies in the last few decades [69]. *R. mucilaginosa* is intrinsically resistant to echinocandins and fluconazole [70,71], as was shown with our isolate EXF-13607, exhibiting high MICs for fluconazole, caspofungin, anidulafungin, and micafungin (256 μg/mL, 32 μg/mL, 32 μg/mL, and 32 μg/mL, respectively). MICs for voriconazole are strain-dependent and usually range between 0.25 μg/mL and 32 μg/mL, with MIC_50_ = 2 μg/mL [69,71]. Considering this, isolate *R. mucilaginosa* EXF-13607 was in vitro susceptible to voriconazole with MIC = 0.032 μg/mL.

The strains *Aureobasidium pullulans* EXF-12432 and *Aureobasidium melanogenum* EXF-13647 were also able to grow at 37 °C. These dematiaceous yeasts, of vast biotechnological importance, were implicated in opportunistic infections in the past [60,72], although later publications suggested that they likely belong to the recently described *A. melanogenum*, which was previously known as *A. pullulans* var. *melanogenum* [73]. Both isolates, EXF-12432 and EXF-13647, showed resistance patterns typical for *A. pullulans*: high fluconazole MICs (MIC_EXF-12432_ = 48 μg/mL and MIC_EXF-13647_ = 16 μg/mL) indicating resistance [74,75].

To the best of our knowledge, apart from *Aureobasidium* spp., *F. magnum*, *N. albida*, and *Rhodotorula* spp., there are no reports on the genera *Articulospora*, *Comoclathris*, *Dothiora*, *Neocucurbitaria*, *Oleoguttula*, *Preussia*, *Pseudogymnoascus*, *Venturia*, *Glaciozyma*, *Mrakia*, *Phenoliferia*, and *Vishniacozyma* being implicated in human infections. Nevertheless, the elevated MICs of *Glaciozyma watsonii* and the *Glaciozyma*-related taxon towards fluconazole (MIC = 256 μg/mL and MIC = 0.25 μg/mL) are noteworthy.

*Penicillium* spp. and *Cladosporium* spp. have their place in clinical mycology and are occasionally reported as agents of infection in immunocompromised patients [76,77]. Like all filamentous fungi, *Penicillium* spp. and *Cladosporium* spp. are resistant to fluconazole [20], with MICs ranging from 16 to 256 μg/mL for our isolates. All three *Penicillium* species tested revealed high in vitro MICs for voriconazole (up to 1.5 μg/mL) and ketoconazole (up to 12 μg/mL). Clinical cases of invasive mycosis caused by voriconazole-resistant *Penicillium* species previously not recognized as primary pathogens have been reported [77]. This is particularly worrying since voriconazole is the first-line therapy for invasive mold infections. Resistance mechanisms responsible for acquired azole resistance in filamentous fungi include point mutations of the drug target gene, overexpression of the target gene and of genes encoding efflux pump proteins, and the formation of biofilm [78]. Those mechanisms have already been observed in *Penicillium digitatum* and are related to the massive use of fungicides in agriculture [78]. Whether the high voriconazole MIC observed in Arctic penicillia is a consequence of acquired resistance and an indication of the dispersion of resistant species from temperate areas to the Arctic or even of evolution of resistant strains in the Arctic is an intriguing question for future studies.

In general, compared to other echinocandins, caspofungin exhibited higher MICs for all tested isolates: four 2-step dilutions higher MIC_50_ than those of anidulafungin and micafungin (0.125 μg/mL for caspofungin versus 0.008 μg/mL for anidulafungin and micafungin). MIC_50_ is usually a species- or strain-specific value and is used here only to facilitate comparisons among different antifungal classes. Such discrepancies are typical of echinocandins. Caspofungin susceptibility is known to show inter- and intra-laboratory variability and should be interpreted in the context of susceptibility to the other two echinocandins [79].

## 5. Conclusions

This study was the first effort to explore potentially pathogenic phenotypes of environmental Arctic fungi. To the best of our knowledge, this includes the first screening for hemolytic ability of Arctic environmental fungal species on blood agar, identifying several species with marked alpha- or beta-hemolytic activity, with one expressing the phenotype at 37 °C and several others able to grow at this temperature despite their polar origins. This is also the first assessment of antifungal susceptibility in Arctic fungi isolated from environments considered to be pristine. We report unexpectedly low susceptibilities for voriconazole in *N. albida* and *Penicillium* spp. and for fluconazole in *Glaciozyma watsonii* and the *Glaciozyma*-related taxon, which should be investigated further to explore their potential clinical implications. The present study contributes to a better understanding of microbiological intrinsic resistance, the development of acquired resistance in environmental isolates, and, most importantly, the emergence of intrinsically resistant species as opportunistic pathogens.

Our research stresses the importance of the identification of virulence-associated phenotypes in Arctic fungi, especially in the context of the increased release of these fungi into the surroundings due to climate change and the recent increment of opportunistic mycoses. The potential hazards these fungi may represent for human and animal health warrant further investigation.

## Figures and Tables

**Figure 1 microorganisms-07-00600-f001:**
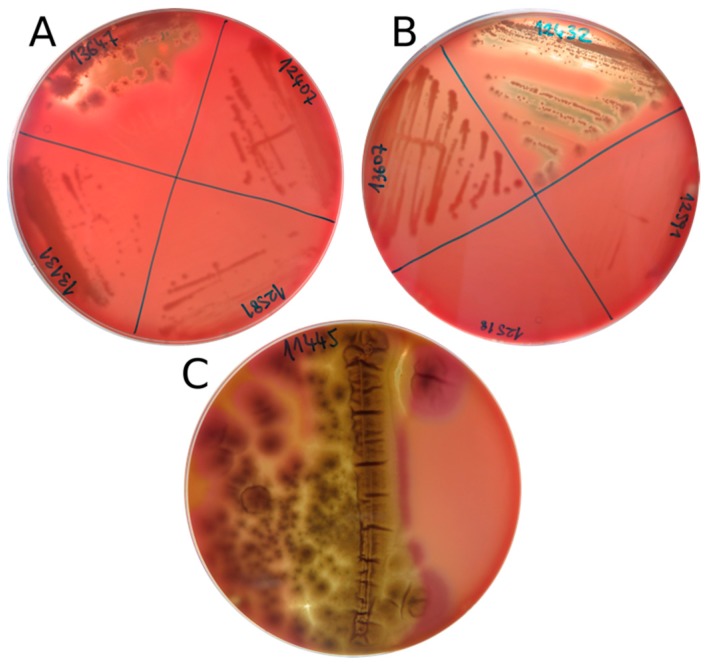
Beta-hemolytic phenotypes of the strains (**A**) *Aureobasidium melanogenum* (EXF-13647), (**B**) *Aureobasidium pullulans* (EXF-12432), and (**C**) *Penicillium bialowiezense*-like (EXF-11445) expressed at 15 °C.

**Table 1 microorganisms-07-00600-t001:** Sampling area and year, sample types, and GPS coordinates of the sampling locations.

Sampling Area	Sampling Location	Sample Type	Sampling Year	GPS Coordinates
Kangerlussuaq, Greenland	Greenland Ice Sheet	Snow	2016/2017	67°04′43″ N 49°20′29″ W
		Dark ice		
		Clear ice		
		Supraglacial water		
	Russell Glacier	Air	2018	67°05′45″ N 50°13′00″ W
Ny-Ålesund, Svalbard	Midtre Lovénbreen	Subglacial ice	2017	78°53′37″ N 12°04′13″ E
		Glacial meltwater		78°53′25″ N 12°03′15″ E
	Vestre Brøggerbreen	Subglacial ice		78°54′55″ N 11°45′48″ E
	Pedersenbreen	Subglacial ice		78°52′46″ N 12°17′57″ E

**Table 2 microorganisms-07-00600-t002:** List of the selected fungal species used in this study, EXF collection number, sample type, sampling location and year, isolation media, and GenBank reference of the deposited sequence used for identification.

Isolate (EXF-)	Species	Phylum	Sample	Sample Location	Sampling Year	Isolation Media *	GenBank Accession Number
11445	*Penicillium bialowiezense*-like	Ascomycota	Dark ice	Greenland Ice Sheet	2016	DG18	MN356456
12407	*Vishniacozyma victoriae*	Basidiomycota	Dark ice	Greenland Ice Sheet	2017	R2A	MK454786
12427	*Cladosporium* sp.	Ascomycota	Dark ice	Greenland Ice Sheet	2017	DG18	MK460331
12432	*Aureobasidium pullulans*	Ascomycota	Snow	Greenland Ice Sheet	2017	SNA	MK460317
12443	*Penicillium chrysogenum*	Ascomycota	Snow	Greenland Ice Sheet	2017	R2A	MK460375
12448	*Preussia* sp.	Ascomycota	Snow	Greenland Ice Sheet	2017	R2A	MK460381
12518	*Mrakia* sp.	Basidiomycota	Dark ice	Greenland Ice Sheet	2017	R2A	MK454832
12523	*Phenoliferia glacialis*	Basidiomycota	Supraglacial water	Greenland Ice Sheet	2017	R2A	MK454852
12580	*Filobasidium magnum*	Basidiomycota	Subglacial ice	Svalbard—Midre Lovenbreen	2017	DRBC	MK670462
12581	*Naganishia albida*	Basidiomycota	Subglacial ice	Svalbard—Midre Lovenbreen	2017	MM	MK670463
12591	*Glaciozyma watsonii*	Basidiomycota	Subglacial ice	Svalbard—Vestre Broggerbreen	2017	SNA	MK670501
12629	*Penicillium crustosum*	Ascomycota	Subglacial ice	Svalbard—Pedersenbreen	2017	MM	MK671620
12639	*Pseudogymnoascus* sp.	Ascomycota	Subglacial ice	Svalbard—Vestre Broggerbreen	2017	SNA	MK671632
12718	*Rhodotorula svalbardensis* pro. tem.	Basidiomycota	Clear ice	Greenland Ice Sheet	2017	DRBC	MK460392
12875	*Comoclathris* sp.	Ascomycota	Snow	Greenland Ice Sheet	2017	R2A	MK460354
12951	*Phenoliferia psychrophenolica*	Basidiomycota	Subglacial ice	Svalbard—Midre Lovenbreen	2017	DG18	MK670452
12990	*Venturia* sp.	Ascomycota	Subglacial ice	Svalbard—Pedersenbreen	2017	R2A	MK671645
13072	*Articulospora* sp.	Ascomycota	Dark ice	Greenland Ice Sheet	2016	DG18	MN356457
13083	*Oleoguttula mirabilis*	Ascomycota	Clear ice	Greenland Ice Sheet	2017	R2A	MK454839
13100	*Neocucurbitaria* sp.	Ascomycota	Snow	Greenland Ice Sheet	2017	SNA	MK460385
13102	*Glaciozyma*-related taxon	Basidiomycota	Glacial meltwater	Svalbard—Midre Lovenbreen	2017	MM	MK670451
13131	*Dothiora* sp.	Ascomycota	Snow	Greenland Ice Sheet	2017	MM	MK460359
13607	*Rhodotorula mucilaginosa*	Basidiomycota	Air	Greenland—Russell glacier	2018	DG18	MN356458
13647	*Aureobasidium melanogenum*	Ascomycota	Air	Greenland—Russell glacier	2018	DG18	MN356459

* Media acronyms stand for: DG18: dichloran 18% glycerol, DRBC: dichloran rose bengal chloramphenicol, MM: minimal medium, SNA: synthetic nutrient-poor agar.

**Table 3 microorganisms-07-00600-t003:** Etest MIC (minimum inhibitory concentration) results for the fungal species tested. MIC values are reported in μg/mL. FL: Fluconazole; CS: Caspofungin; VO: Voriconazole; MYC: Micafungin; KE: Ketoconazole; AND: Anidulafungin. NG: no growth; NT: not tested (assumed to be intrinsically resistant).

EXF-	Species	Phylum	MIC (μg/mL)					
			FL	CS	VO	MYC	KE	AND
13072	*Articulospora* sp.	Ascomycota	256	0.032	0.004	0.012	0.002	0.023
13647	*Aureobasidium melanogenum*	Ascomycota	16	0.25	0.38	0.25	0.5	0.125
12432	*Aureobasidium pullulans*	Ascomycota	48	0.38	0.094	0.094	0.19	0.38
14635 CTRL	*Candida albicans*	Ascomycota	0.032	0.064	≤0.002	0.016	≤0.002	0.008
12427	*Cladosporium* sp.	Ascomycota	16	2	0.023	0.008	0.006	0.004
12875	*Comoclathris* sp.	Ascomycota	256	0.125	0.38	0.064	0.5	0.006
13131	*Dothiora* sp.	Ascomycota	0.38	0.064	0.003	0.006	0.003	0.002
13100	*Neocucurbitaria* sp.	Ascomycota	256	32	0.002	0.006	0.004	0.002
13083	*Oleoguttula mirabilis*	Ascomycota	256	0.002	0.002	0.002	0.002	0.002
11445	*Penicillium bialowiezense*-like	Ascomycota	256	0.016	1.5	0.002	0.38	0.002
12443	*Penicillium chrysogenum*	Ascomycota	256	0.016	0.5	0.002	2	0.002
12629	*Penicillium crustosum*	Ascomycota	256	0.5	0.25	0.008	12	0.008
12448	*Preussia* sp.	Ascomycota	256	0.125	0.19	0.004	0.032	0.016
12639	*Pseudogymnoascus* sp.	Ascomycota	8	1	0.094	0.125	0.38	0.25
12990	*Venturia* sp.	Ascomycota	64	0.002	0.016	0.125	0.032	0.032
12580	*Filobasidium magnum*	Basidiomycota	4	NT	0.032	NT	0.023	NT
13102	*Glaciozyma*-related taxon	Basidiomycota	0.25	NT	0.002	NT	0.002	NT
12591	*Glaciozyma watsonii*	Basidiomycota	256	NT	0.002	NT	0.002	NT
12518	*Mrakia* sp.	Basidiomycota	NG	NG	NG	NG	NG	NG
12581	*Naganishia albida*	Basidiomycota	64	NT	2	NT	0.047	NT
12523	*Phenoliferia glacialis*	Basidiomycota	0.016	NT	0.002	NT	0.004	NT
12951	*Phenoliferia psychrophenolica*	Basidiomycota	0.047	R	0.002	R	0.002	R
12718	*Rhodotorula svalbardensis* pro. tem.	Basidiomycota	NG	NG	NG	NG	NG	NG
13607	*Rhodotorula mucilaginosa*	Basidiomycota	256	32	0.032	32	0.032	32
12407	*Vishniacozyma victoriae*	Basidiomycota	1	NT	0.016	NT	0.032	NT
Range			0.016–256	0.002–32	0.002–2	0.002–32	0.002–12	0.002–32
Geometric mean MIC			114.8	0.263	0.672	4.57	2.24	2.35
MIC_50_			64	0.032	0.032	0.125	0.008	0.008
MIC_90_			256	0.5	0.5	32	0.25	0.5

**Table 4 microorganisms-07-00600-t004:** Hemolytic phenotypes of the fungal species tested.

EXF-	Species	Phylum	Hemolysis in Bovine Blood	
			15 °C	37 °C
11445	*Penicillium bialowiezense*-like	Ascomycota	α/β −	NG
12407	*Vishniacozyma victoriae*	Basidiomycota	γ	NG
12427	*Cladosporium* sp.	Ascomycota	γ	NG
12432	*Aureobasidium pullulans*	Ascomycota	β	NG
12443	*Penicillium chrysogenum*	Ascomycota	α −	γ − (little growth)
12448	*Preussia* sp.	Ascomycota	α	NG
12518	*Mrakia* sp.	Basidiomycota	γ	NG
12523	*Phenoliferia glacialis*	Basidiomycota	γ	NG
12580	*Filobasidium magnum*	Basidiomycota	γ	NG
12581	*Naganishia albida*	Basidiomycota	γ	γ − (little growth)
12591	*Glaciozyma watsonii*	Basidiomycota	γ	NG
12629	*Penicillium crustosum*	Ascomycota	γ	NG
12639	*Pseudogymnoascus* sp.	Ascomycota	α *	NG
12718	*Rhodotorula svalbardensis* pro. tem.	Basidiomycota	NG	NG
12875	*Comoclathris* sp.	Ascomycota	γ	NG
12951	*Phenoliferia psychrophenolica*	Basidiomycota	α −	NG
12990	*Venturia* sp.	Ascomycota	γ	NG
13072	*Articulospora* sp.	Ascomycota	γ	NG
13083	*Oleoguttula mirabilis*	Ascomycota	α	NG
13100	*Neocucurbitaria* sp.	Ascomycota	α -	NG
13102	*Glaciozyma*-related taxon	Basidiomycota	γ	NG
13131	*Dothiora* sp.	Ascomycota	γ	NG
13607	*Rhodotorula mucilaginosa*	Basidiomycota	α -	β
13647	*Aureobasidium melanogenum*	Ascomycota	β	γ − (little growth)
14635 CTRL	*Candida albicans*	Ascomycota	γ	β

α: alpha-hemolytic, β: beta-hemolytic, γ: gamma-hemolytic, + signifies a stronger phenotype, − signifies a weaker phenotype, NG: no growth. * α-hemolysis is noted for this isolate even though the phenotype does not correspond to the exact description of an α-hemolytic organism.

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
