# Peer review of "Phenotypes Associated with Pathogenicity: Their Expression in Arctic Fungal Isolates"

_microorganisms, 2019, doi:10.3390/microorganisms7120600_

Round 1

Reviewer 1 Report

The work by Perini and colleagues is on an actual and important topic that brings together climate changes and emergence of potential pathogens. This topic is relevant if we consider the fact that, specially in developed countries, elderly population is increasing with all the implications it has for public health.

The study was well designed and executed and is interesting for different readers. Nevertheless, there are typing and formatting issues through the document that must be corrected. In my opinion, table 1 should be formatted in such a way that would be easy to identify the sample types collected from each location. As it is now, at least for me, it is not clear. I would also suggest that the last paragraph of the conclusion would be modified in order to strengthen the conclusions.

Author Response

We would like to thank the reviewers for their very constructive and helpful remarks. We attempted to follow the suggestions in full. The detailed answers to the raised questions are listed below.

REVIEWER 1

COMMENT: The study was well designed and executed and is interesting for different readers. Nevertheless, there are typing and formatting issues through the document that must be corrected. In my opinion, table 1 should be formatted in such a way that would be easy to identify the sample types collected from each location. As it is now, at least for me, it is not clear.

ANSWER: The format of Table 1 has been changed to help with the visualization of locations and sample types collected as requested.

COMMENT: I would also suggest that the last paragraph of the conclusion would be modified in order to strengthen the conclusions.

ANSWER: The last paragraph of the conclusions has been strengthen as suggested (lines 380-383):

“Our research stresses the importance of the identification of virulence-associated phenotypes in Arctic fungi, especially in the context of the increased release of these fungi into the surroundings due to climate change and of the recent increment of opportunistic mycoses. The potential hazards these fungi may represent for the human (and animal) health warrant further investigation.”

Reviewer 2 Report

This paper describes the isolation of fungal species from the Arctic region and exploring their growth at 37°C, expression of hemolytic activity on bovine blood agar plates and resistance to common antifungal therapeutics in a satisfactory way. The work is interesting, informative and well-presented. However, I have some minor observations that are listed in the following lines.

The manuscript would benefit from close editing. Lines 117-126: Add what the media abbreviations stand for. Line 145: fungal isolates Line 152: Prior to the test Line 154: was prepared by homogenizing Line 171: … (147/320), …(172/320), and … (1/320): convert to percentages. Line 182: Add what the media abbreviations stand for in the caption of Table 2. Line 188: … able to grow at…

Author Response

We would like to thank the reviewers for their very constructive and helpful remarks. We attempted to follow the suggestions in full. The detailed answers to the raised questions are listed below.

REVIEWER 2

COMMENT: The manuscript would benefit from close editing. Lines 117-126: Add what the media abbreviations stand for.

ANSWER: The media abbreviations have been added as suggested.

COMMENT: Line 145: fungal isolates

ANSWER: Amended.

COMMENT: Line 152: Prior to the test

ANSWER: Amended.

COMMENT: Line 154: was prepared by homogenizing

ANSWER: Amended.

COMMENT: Line 171: … (147/320), …(172/320), and … (1/320): convert to percentages.

ANSWER: Numbers have been converted to percentages as suggested.

COMMENT: Line 182: Add what the media abbreviations stand for in the caption of Table 2.

ANSWER: Media abbreviations have been added.

COMMENT: Line 188: … able to grow at… 

ANSWER: Amended.